# Mechanical and Biological Characterization of PMMA/Al_2_O_3_ Composites for Dental Implant Abutments

**DOI:** 10.3390/polym15153186

**Published:** 2023-07-27

**Authors:** Ilaria Roato, Tullio Genova, Donatella Duraccio, Federico Alessandro Ruffinatti, Diletta Zanin Venturini, Mattia Di Maro, Alessandro Mosca Balma, Riccardo Pedraza, Sara Petrillo, Giorgia Chinigò, Luca Munaron, Giulio Malucelli, Maria Giulia Faga, Federico Mussano

**Affiliations:** 1CIR Dental School, Department of Surgical Sciences, University of Turin, Via Nizza 230, 10126 Torino, Italy; ilaria.roato@unito.it (I.R.); alessandro.moscabalma@unito.it (A.M.B.); riccardo.pedraza@polito.it (R.P.); federico.mussano@unito.it (F.M.); 2Department of Life Sciences and Systems Biology, University of Turin, Via Accademia Albertina 13, 10123 Torino, Italy; tullio.genova@unito.it (T.G.); federicoalessandro.ruffinatti@unito.it (F.A.R.); diletta.zanin@edu.unito.it (D.Z.V.); giorgia.chinigo@unito.it (G.C.); luca.munaron@unito.it (L.M.); 3Institute of Sciences and Technologies for Sustainable Energy and Mobility, National Council of Research, Strada delle Cacce 73, 10135 Torino, Italy; mattia.dimaro@stems.cnr.it (M.D.M.); mariagiulia.faga@stems.cnr.it (M.G.F.); 4Department of Molecular Biotechnology and Health Sciences, Molecular Biotechnology Center (MBC), University of Turin, Via Nizza 52, 10126 Torino, Italy; sara.petrillo@unito.it; 5Politecnico di Torino, Department of Applied Science and Technology, C.so Duca Degli Abruzzi 24, 10129 Torino, Italy; giulio.malucelli@polito.it

**Keywords:** PMMA composites, alumina, flexural modulus, viability, cell adhesion, osteoconductivity

## Abstract

The mechanical and biological behaviors of PMMA/Al_2_O_3_ composites incorporating 30 wt.%, 40 wt.%, and 50 wt.% of Al_2_O_3_ were thoroughly characterized as regards to their possible application in implant-supported prostheses. The Al_2_O_3_ particles accounted for an increase in the flexural modulus of PMMA. The highest value was recorded for the composite containing 40 wt.% Al_2_O_3_ (4.50 GPa), which was about 18% higher than that of its unfilled counterpart (3.86 GPa). The Al_2_O_3_ particles caused a decrease in the flexural strength of the composites, due to the presence of filler aggregates and voids, though it was still satisfactory for the intended application. The roughness (Ra) and water contact angle had the same trend, ranging from 1.94 µm and 77.2° for unfilled PMMA to 2.45 µm and 105.8° for the composite containing the highest alumina loading, respectively, hence influencing both the protein adsorption and cell adhesion. No cytotoxic effects were found, confirming that all the specimens are biocompatible and capable of sustaining cell growth and proliferation, without remarkable differences at 24 and 48 h. Finally, Al_2_O_3_ was able to cause strong cell responses (cell orientation), thus guiding the tissue formation in contact with the composite itself and not enhancing its osteoconductive properties, supporting the PMMA composite’s usage in the envisaged application.

## 1. Introduction

Dental implants have become the therapeutic choice for replacing missing teeth over the last few decades. They provide a sturdy and long-lasting solution that closely mimics the appearance and function of natural teeth. Implants also help prevent bone loss in the jaw, which may occur when a tooth is missing, owing to the reliability of osseointegration, i.e., the direct deposition of new bone adjacent to the fixture [1]. According to many works in the scientific literature, titanium-roughened surfaces enhance osseointegration [2] and are, therefore, available in the majority of marketed implant systems. Nevertheless, some researchers have pointed out the release of titanium ions/particles within adjacent tissues [3] as a result of tribocorrosion [4] or implantoplasty [5] and their possible role in promoting allergies, which may be underestimated due to the detection protocols being optimized for dermatology [6].

Aimed at avoiding, or at least limiting, the use of titanium and its alloys in dental implant systems, advanced massive oxidic ceramics have been proposed in the last few years. For example, zirconia and alumina-toughened zirconia (ATZ) have been used for dental prosthetics as possible substitutes for titanium. This is because their restorations have better esthetics both due to their excellent optical properties and to the absence of the black line that is caused by metal in the cervical line of traditional restorations [7,8]. However, in the face of masticatory forces, zirconia and ATZ abutments can develop surface defects and promote plastic deformation in the metal fixture [9]. Moreover, there is still a huge concern about the long-term durability of these materials due to the phase transition towards the monoclinic phase (ageing process), which can limit their application in the biomedical field [10]. A growing number of clinicians are therefore oriented toward hybrid solutions that combine titanium-based intraosseous fixtures with polymers for the fabrication of intra-oral prosthetic components, i.e., fixed dental prosthesis (FDP) frameworks, abutment screws, prosthetic implant abutments, and clip-on implant bars [11].

Indeed, polymer materials display a series of advantages over classic metal–ceramic prosthetic frameworks, such as having a more favorable elongation to fracture (compared to that of ceramics) and being cost effective and lightweight compared to ATZ and ceramics in general [12]. Moreover, polymer materials can be easily integrated into the digital workflow [13], which is becoming strategic in the dental industry [14]. Among the most promising candidates suitable for implant-supported dental prostheses, polyetheretherketone (PEEK) [11] and polymethylmethacrylate (PMMA) should be considered [15]. The former has been gaining a lot of interest even as an intraosseous material to fabricate fixtures [16] and provisional dental prostheses [17], but it is hindered in applications as a prosthetic by its inertness, which severely limits adhesive bonding [18]. PMMA has already been used for orthopedic cements, demonstrating good bio-compatibility [19]. As it is already widely used for removable prostheses due to its aesthetics, ease of repair, and reduced cost, it seems the most promising solution for dental implant abutments and frameworks as well [20,21]. However, due to having a lower flexural strength with respect to that of other materials, it is more suitable for short-term use or temporary applications [22].

PMMA/alumina (Al_2_O_3_) composites were introduced successfully more than two decades ago [23]. Nevertheless, the chemical incompatibility of Al_2_O_3_ with the organic matrix along with the tendency of nano- and micro-particles to agglomerate have prompted a series of fabrication approaches to achieve the uniform distribution of alumina throughout the PMMA matrix [24,25]. Moreover, one of the major problems that may have hindered the clinical application of these composites is their long-term stability. It is well known that the salivary environment can lead to the degradation of PMMA by increasing the diffusion of residual MMA monomers [26].

To adapt the features of PMMA to different clinical requirements, various fillers have been recently introduced to prepare PMMA composites, attaining improved biocompatibility [21] or antimicrobial properties [27]. Less studied, however, is the choice of the best composite material suitable for the gingival interface (such as prosthetic implant abutments), which requires, at the same time, a high mechanical modulus and the ability to interact optimally with oral soft tissues, establishing and maintaining the so-called mucosal seal. Noteworthy in this context is the study by McKinney et al. [28], who described ultrastructurally the formation of hemidesmosomes at the interface between epithelial cells and alumina implants.

With this aim, we propose and characterize three different formulations of PMMA/Al_2_O_3_ composites with regard to their possible application in implant-supported prostheses, taking into consideration both their mechanical and biological characteristics. Further, for preparing PMMA and its composites, we have used an isostatic press in a dedicated chamber at a pressure between six and eighth bars and a temperature of 75 °C for 3 h, in order to complete the curing and reduce the presence of the unreacted monomer.

## 2. Materials and Methods

### 2.1. Material

Sub-micrometric alumina powder with an average particle size of 3 µm, a specific surface area of 3 m^2^/g, and purity of 99.9% was purchased from 2B Minerals S.r.l. (Campogalliano, Modena, Italy). Methyl methacrylate (MMA) and benzoyl peroxide (employed as radical initiator) were purchased from Aldrich (Milano, Italy) and used without further purification.

### 2.2. Preparation of the PMMA/Al_2_O_3_ Composites

MMA, radical initiator, and sub-micrometric alumina powder were mechanically mixed for 24 h in a climate-controlled room at a temperature between 23 and 30 °C. Then, the mixture was compressed using an isostatic press in a dedicated chamber at a pressure between 6 and 8 bars and a temperature of 75 °C for 3 h, in order to complete the curing and reduce the presence of unreacted monomer. Three different formulations were obtained: PMMA/Al_2_O_3_ 70/30, PMMA/Al_2_O_3_ 60/40, and PMMA/Al_2_O_3_ 50/50 (the amounts are intended to be the weight%). PMMA alone was also prepared as reference material. A 50 mm × 10 mm × 3 mm rectangular-shaped mold was used. For biological tests, samples were compressed into cylindrical specimens (with a diameter of 12 mm and height of 0.8 mm).

### 2.3. Morphological Analysis

The surface morphology of PMMA-based composites was investigated by scanning electron microscopy (SEM) using a ZEISS EVO 50 XVP (Oberkochen, Germany), equipped with a LaB_6_ source. The distribution of the alumina on the surface of the materials was evaluated by EDX (Oxford INCA Energy 200, Oxford Instrument, Abingdon, Oxfordshire, UK). To minimize the charge effects, the surface of the materials was previously coated with a thin chromium layer (∼10 nm). Further SEM images and EDX analyses were carried out on PMMA composites after ASCs cultured in osteogenic medium for 30 days (see Section 2.11) in order to verify the calcification.

### 2.4. WAXD Analysis

XRD patterns of PMMA and its composites were acquired by using a PANalytical PW3040/60 X’Pert PRO MPD diffractometer (Malvern, UK) in Bragg−Brentano geometry and using a Ni-filtered Cu Kα radiation (λ = 0.15418 nm), generated at 45 kV and 40 mA. WAXD profiles were obtained with a continuous scan of 0.04°/s (scan step size: 0.0167°; time per step: 53 s) in the range of 5–80°.

### 2.5. Mechanical Properties

The Shore D hardness of PMMA and its composites was measured using a Sauter Classic Durometer (Sauter, Wutoschingen, Germany), according to ASTM D2240 [29]. The analysis was performed on five different specimens for each composition, and the results were averaged.

Flexural modulus and flexural strength were measured by three-point bending tests performed on an Instron universal testing machine (Instron 5966 dynamometer, Norwood, MA, USA) according to the ASTM D790-03 standard [30]. The sample size was 50 mm × 10 mm × 3 mm. The span to depth ratio was about 30 and the cross-head speed was maintained at 1.1 mm/min until fracture occurred. For each material, ten specimens were analyzed, and the results were averaged.

### 2.6. Surface Roughness and Wettability

Surface micro-roughness was evaluated using Form Talysurf 120 contact profilometer (Taylor-Hobson, New Star Road, UK) equipped with a 2 μm diamond conical stylus. In accordance with ISO 4288 [31], the parameter R_a_, which corresponds to the arithmetic mean deviation of the assessed profile, was measured, keeping the length of the measurement at 2.4 mm and using a cut-off of 0.8 mm.

The influence of the Al_2_O_3_ on the surface wettability of PMMA was investigated by means of water contact angle measurements, performed with a Theta Lite Optical Tensiometer (Biolin Scientific, Stockholm, Sweden). The contact angle was measured through the sessile drop method, using double-distilled water as liquid phase at RT. At least 5 different measurements were carried out for each sample, and the results were averaged.

### 2.7. Protein Adsorption Assay

A low-protein concentration solution (2 wt.% of fetal bovine serum (FBS, Life Technologies, Milan, Italy)) in phosphate-buffered saline (PBS)) was utilized to incubate the PMMA/Al_2_O_3_ composite disks at 37 °C for 20 min. Subsequently, the samples were washed twice with PBS, and the adsorbed proteins were eluted from the disks with Tris Triton buffer (10 mM Tris (pH 7.4), 100 mM NaCl, 1 mM EDTA, 1 mM EGTA, 1% Triton X-100, 10% glycerol, and 0.1% SDS) for 10 min. Finally, the total protein amount was quantified with a PierceTM BCA Protein Assay Kit (Life Technologies, Carlsbad, CA, USA) following the manufacturer’s instructions.

### 2.8. Cell Culture

To characterize the biological response in vitro, adipose stem cells ASC52hTert (ATCC), human fibroblasts HFF (NHDF, ECACC, Salisbury, UK), and keratinocytes HaCaT (ATCC) cell lines were maintained in Alpha-MEM (Life Technologies, Milano, Italy) with 10% FBS, 100 U/mL of penicillin, and 100 μg/mL of streptomycin. Cells were passaged at sub-confluence to prevent contact inhibition and were kept under a humidified atmosphere of 5% CO_2_ at 37 °C.

### 2.9. Cell Adhesion and Viability Assays

For cell adhesion, composites were placed in 48-well plates (BD, Franklin Lakes, NJ, USA), and, on the surface of each disk, 7000 cells were seeded in 70 µL of growth medium after being detached with trypsin and carefully counted. The 48-well plates were kept at 37 °C with 5% CO_2_ for 15 min. Samples were washed twice with PBS to eliminate non-adherent cells; the adherent cell amount was assessed by Cell Titer GLO (Promega, Madison, WI, USA) according to the manufacturer’s protocol.

To assess cell viability, cells were plated at a density of 2500 cells in 70 µL of growth medium on each disk surface and placed in 48-well culture dishes. Viability was assessed by Cell Titer GLO (Promega) according to the manufacturer’s protocol at 24 h, 48 h, and 72 h.

### 2.10. Cell Morphology and Orientation Analysis

Cells were seeded on the samples at a concentration of 7000 cells/sample in a 48-well plate (BD) and then kept in growth conditions. After 1 and 24 h, the specimens were washed in PBS, before fixing the cells with 4% paraformaldehyde in PBS for 10 min. After being washed with PBS, cells were permeabilized with 0.1% Triton X-100 (Sigma-Aldrich, Milano, Italy) in PBS. Cells were stained with Alexa 488-Phalloidin (Life Technologies, Milano, Italy) to detect the cytoskeleton. Images were acquired with a Nikon Eclipse Ti-E microscope using different objectives: Nikon Plan 10X/0.10; Nikon Plan Fluor 40X/0.75; and Nikon Plan Apo VC 60X/1.40 (Nikon Instruments, Amsterdam, The Netherlands).

A directionality analysis was performed using an automated software developed in our laboratory called MORPHEUS. The tool was used according to the workflow reported in the literature [32].

### 2.11. Real-Time qRT-PCR

ASC52hTert cells were grown on PMMA-based composites in osteogenic medium (OM) containing Alpha-MEM, supplemented with 10% FBS, 50 μg/mL ascorbic acid, 10^−8^ M dexamethasone, and 10 mM beta-glycerophosphate (Sigma-Aldrich, Milano, Italy). After 30 days, cells were detached by Trypsin/EDTA treatment, and subsequently washed and dissolved in TRIzol reagent (Thermofisher, Waltham, MA, USA) for RNA extraction. One microgram of RNA was converted up to single-stranded cDNA by the High-Capacity cDNA Reverse Transcription Kit (Applied Biosystems). The mRNA expression of the following genes was tested: alkaline phosphatase (ALP, NM_000478.5), collagen 1 (COLL-1, NM_000088.3), and osteocalcin (OCN, NM_199173.5); the primer sequences were published previously [33]. RT-PCR was performed with Luna^®^ Universal qPCR Master Mix (New England BioLabs, Ipswich, MA, USA), using the CFX96 system (Bio-Rad, Hercules, CA, USA). The amplification protocol had 40 cycles with a Tm of 58° C. The expression of β-actin was chosen to normalize gene expression data, and the 2^−ΔΔC^_T_ method was used for the quantitative analysis with CFX Maestro 1.0 Manager software (Bio-Rad, Hercules, CA, USA).

### 2.12. Statistical Analysis

A descriptive analysis was performed with the data presented via mean ± standard error of mean (±SEM). Data were first tested for normality using a Shapiro–Wilk test, then a multiple comparison was performed with an ordinary one-way ANOVA. In case the data collected were nonparametric, differences among groups were analyzed with a Kruskal–Wallis test. All data were analyzed by means of Graph/pad/Prism 4 software (GraphPad Software, Inc., La Jolla, CA, USA). All statistical comparisons were conducted with a 0.05 level of significance. Orientation analysis was performed considering a circular statistic. The significance of directionality distributions was tested with both Rayleigh test and V-test. The former allowed us to assess whether a privileged direction was present, while the latter identified which specific direction was preferred. At the populational level, differences were assessed with a two-way ANOVA.

## 3. Results

### 3.1. Morphological Analysis

The typical SEM micrographs of the surfaces of unfilled PMMA and its composite are reported in Figure 1. The unfilled PMMA (Figure 1A) shows a fairly smooth and compact surface due to the use of an isostatic press during the polymerization process. Alumina particles are visible and appear to be not well embedded in the matrix in the composites (micrographs in Figure 1B–D). Although some aggregates of the particles are visible in all of the composites, they increase in number and size as the amount of Al_2_O_3_ increases. Furthermore, the presence of alumina, reducing PMMA chain mobility and thus leading to an increase in the viscosity of the mixture during curing [34,35,36], causes the formation of holes throughout the composites. Again, the number of holes increases as the amount of alumina increases. For these reasons, the sample with the highest alumina content (i.e., 50 wt.%) shows an extremely irregular surface consisting of large and numerous agglomerates of particles and holes (Figure 1D), with particles trapped within the holes.

### 3.2. Structural Analysis

Figure 2 collects the WAXD profiles of PMMA and its composites. PMMA is an amorphous polymer with a diffraction pattern characterized by a broad halo at about 2θ = 13° linked to the interchain local distances and other two halos due to the ordering inside the main polymer chains (i.e., sindio, iso, or eterotactic units) [37]. The composite spectra are characterized by the presence of the most intense reflections of the rhombohedral structure of α-alumina [38] and by the absence of a halo at 2θ = 13°. This result indicates that the presence of the filler increases the disorder and the distance among the PMMA polymer chains.

### 3.3. Mechanical Properties

The effect of Al_2_O_3_ inclusions on the hardness of PMMA-based composites is reported in Table 1. The addition of alumina particles slightly enhances the Shore D hardness of the polymer from 91.8 to 96.8 (for the PMMA/Al_2_O_3_ 70/30 composite). By further increasing the alumina loading in the composites, no significant increase in PMMA hardness is found. This is probably due to the fact that, at very high alumina contents (i.e., 40 and 50 wt.%), the hardening effect promoted by the hard Al_2_O_3_ filler is counterbalanced by softening due to the high numbers of voids. Table 1 also collects the values of the flexural parameters for PMMA and its composites. The presence of Al_2_O_3_ particles induces an increase in the flexural modulus of the polymer matrix. In particular, the flexural modulus of PMMA containing 40 wt.% Al_2_O_3_ is about 18% higher than that of its unfilled counterpart. Thus, the higher the flexural modulus, the less easily the material is deformed. Therefore, PMMA composites may better suppress the deformation and fracturing of the denture under occlusal pressure compared to unfilled PMMA [39]. Conversely, with the addition of alumina, filler aggregates and inner defects (i.e., voids) are easily introduced in the composite system, as confirmed by our SEM observations (Figure 1B–D), hence decreasing the PMMA flexural strength [40], though still satisfactory for short-term use or temporary applications.

### 3.4. Surface Properties (Wettability and Roughness)

The effect of alumina on the wettability of PMMA can be clearly observed by contact-angle measurements (Table 1). According to the literature [41], PMMA is considered a hydrophilic polymer, as its contact angle with water is about 77.2°, i.e., a lot lower than 90°. The addition of 30 wt.% alumina to the polymer matrix increases the water contact angle to 99.7°, hence overcoming the threshold limit value of 90° and making the polymer hydrophobic. By further increasing the Al_2_O_3_ loading, the composites become more hydrophobic, reaching the value of 105.8° in the case of the PMMA/Al_2_O_3_ 50/50 composite. The difference in the water contact angle between PMMA and the PMMA/Al_2_O_3_ 50/50 composite is highlighted in Figure 3. This result is unexpected considering that (a) alumina is deemed inherently hydrophilic, and (b) fillers such as TiO_2_ and ZrO_2_ are expected to improve the hydrophilic behavior of the polymer matrix surface [42,43]. However, in our previous work [33], in which we analyzed the role of alumina-toughened zirconia (ATZ) loading on the mechanical and biological properties of UHMWPE, we found a contact angle of 119° for UHMWPE/ATZ 97.5/2.5 wt.% (whereas the UHMWPE contact angle with water was about 86°). One possible explanation for these results could be attributed to the changes (particularly the increase) in the surface roughness of PMMA-based composites, as described below.

The roughness (R_a_), shown in Table 1, follows the same trend as the wettability and increases from 1.94 µm for unfilled PMMA to 2.45 µm for PMMA/Al_2_O_3_ 50/50. This result indicates that the surface roughness influences the surface wettability. This strong correlation, however, is not easy to explain, as several possible contributions may play a role, such as the distribution of filler on the surface [40].

### 3.5. Protein Adsorption

Figure 4 reports the protein adsorption, which is a process occurring whenever a material is put in contact with bodily fluids (such as saliva or blood), for the unfilled PMMA and PMMA composites. It is clear that by increasing the wt.% of alumina, the protein adsorption significantly increases and is even doubled for PMMA/Al_2_O_3_ 60/40 (C) and PMMA/Al_2_O_3_ 50/50 with respect to that of the unfilled polymer. This means that the alumina plays a crucial role in the protein adsorption, which is indeed the first process that occurs after implantation. Surface characteristics influence protein adsorption, depending on the type of protein; as stated by Wei et al. [44], albumin, a major component of serum, has shown greater adsorption on hydrophobic surfaces, in agreement with the current data, while fibronectin, in contrast, has shown hydrophilic behavior. Protein adsorption drives cell adhesion on alumina [45] and may influence gingival cells [46].

### 3.6. Cell Adhesion and Viability

To test how the presence of alumina affects the first stages of cell–material interactions, cell adhesion assays were carried out (Figure 5). Interestingly, the presence of alumina in the composites significantly improved adhesion both in stem and differentiated cells with respect to that found for the unfilled PMMA. The differences among the PMMA/Al_2_O_3_ composites are negligible for ASCs, while it seems noteworthy that PMMA/Al_2_O_3_ 70/30 does not outperform the unfilled PMMA in HaCaT cells. A possible explanation for this behavior may be due to the several sets of integrins operating the surface recognition, characterizing epithelial cells (keratinocytes) vs. stromal cells (mesenchymal stem cells and fibroblasts) [47].

The cell viability tests performed on the four specimens confirmed that the materials are all biocompatible (no cytotoxicity effects were found) and able of sustaining cell growth and proliferation (Figure 6). At 24 and 48 h, the composites behaved like the unfilled PMMA for both stem and differentiated cells. Conversely, at 72 h, the ASCs grew significantly more on the unfilled PMMA and PMMA/Al_2_O_3_ 70/30 than on the other two PMMA/Al_2_O_3_ composites, in agreement with the study by Chiang et al. [48], in which the increase in the reinforcement (in their study, the calcium sulphate was over the limit of 10 wt.%) within the PMMA matrix was not followed by an analogous enhancement in cell proliferation. Interestingly, keratinocytes and fibroblasts did not replicate the growth pattern of the ASCs, but proliferated more on the unfilled PMMA, PMMA/Al_2_O_3_ 60/40, and (only the fibroblasts) PMMA/Al_2_O_3_ 50/50. Quite peculiarly, PMMA/Al_2_O_3_ 70/30 reduced, in a statistically significant way, cell proliferation in both HaCaT and NHDF. Moreover, the different surface patterns of the composites could directly interact with the proper cell type’s ability to distribute on them.

### 3.7. Cell morphology and Orientation Analysis

The morphological features of the fibroblasts on the PMMA and PMMA composites at 1 (Figure 7) and 24 h (Figure 8) were investigated to see whether differences in the composition affected their cytoskeleton organization and orientation. Among the soft tissue cells studied, fibroblasts were selected as the most suitable model, as HaCaT cells are quite difficult to obtain in single-cell culture. Figure 7 shows that 1 h after plating, some cells are still hemispherical with fewer and reduced lamellipodia/filopodia, whereas others appear to have spread with evident cellular protrusions. In particular, the cells on the PMMA composites (Figure 7B–D) display larger surfaces than those plated on the unfilled PMMA. Indeed, they present a more spread out and extended shape that results in a higher occupied area. Interestingly, the fibroblasts on PMMA/Al_2_O_3_ 60/40 (Figure 7C) are more elongated than those on PMMA/Al_2_O_3_ 70/30 (Figure 7B), with less broad lamellipodia and more evident filopodia oriented toward a specific direction. In the PMMA/Al_2_O_3_ 50/50 composite (Figure 7D), directionality is still partially conserved but the broad lamellipodia are recovered and the filopodia are thinner. Furthermore, as depicted in Figure 8, all the materials presented well-developed cells evenly distributed throughout the sample at 24 h. The cells on PMMA (Figure 8A) show a spread-out flat cell body with long protruding cytoplasmic extensions. In the PMMA/Al_2_O_3_ 70/30 composite (Figure 8B), the protruding extensions are less evident, and the cell area is more extended. The fibroblasts on PMMA/Al_2_O_3_ 60/40 (Figure 8C) exhibit a more elongated and spindle-shaped morphology with thin and long cytoplasmic protrusions, confirming the trend stated for cells fixed at 1 h. Similarly, directionality is also evident at 24 h for the PMMA/Al_2_O_3_ 50/50 composite (Figure 8D), in accordance with previous findings, with the cells elongated but with a more spread-out morphology and quite directional filopodia.

A cell orientation analysis was performed on the same dataset of fluorescent images using the specific automated software MORPHEUS (1.0) [32]. At both 1 h and 24 h from seeding, PMMA/Al_2_O_3_ 60/40 showed the highest degree of orientation, confirming our fluorescence microscopy observations. MORPHEUS calculated the orientation distribution at a cellular scale for each picture, considering a range of 180 ° (from +90° to −90°). Based on these distributions, a significant directionality in the samples was observed. In particular, it was possible to assess whether the sample was uniformly distributed around a half circle and if it had a common mean direction. The latter was directly related to the concept of “steering power”, to refer to and emphasize a potentially privileged cell direction. The steering power was measured with the length of the main resultant vector R of the distribution obtained. The *p*-values from the Rayleigh test and V-test against the mean direction are shown in Table 2.

In addition, to assess the presence of a steering effect at the populational level, a two-way ANOVA was performed on the R values grouped by the culture time and amount of alumina in the composites (Figure 9). The analysis showed no significant interaction between the two variables, indicating that the alumina-induced orientation effect has the same trend at both 1 h and 24 h (Figure 9). However, 24 h of culture systematically showed an increase in orientation power (the main effect of culture time, *p* value was 6.93 × 10^−4^).

At the population level, the overall data confirmed that the directionality induced by the alumina was at the maximum at 40 wt.% loading. Interestingly, it did not follow a monotonous trend, providing a lower but still significant steering power when the amount of alumina was 50 wt.%. These changes in cell morphology are likely due to the combined effect of the surface topology, morphology, and chemical composition of the composites. The analysis of the cell orientation, however, showed that the presence of alumina in PMMA results in specific cell responses, thus driving tissue formation in contact with the composite itself. This could be beneficiary in the field of tissue engineering, which aims at the (re)generation of new and functional tissues.

### 3.8. Alumina Filler Decreases Osteogenic Genes and Mineralization

Both the unfilled PMMA and its composites were tested for their possible osteoconductivity by evaluating the induction of three representative osteogenic genes (alkaline phosphatase—ALP, collagen type I—COLL I, and osteocalcin—OCN) in the ASCs after 30 days of culture. The charts in Figure 10 show that PMMA/Al_2_O_3_ 70/30 is consistently less osteogenic than the unfilled PMMA for all the genes. Less evident differences characterize PMMA/Al_2_O_3_ 60/40, which has lower values than the unfilled polymer, but never in a statistically significant way. Further, the behavior of PMMA/Al_2_O_3_ 50/50 is between that of the other two composites. This gene profile is consistent with the SEM images reported in Figure 11 obtained on ASCs cultured in the same conditions as the qPCR, showing a calcification that is remarkably greater on the unfilled PMMA (the calcium content was also evaluated through EDX as reported in the inset of Figure 11) than on the three composites (Figure 11B–D).

Finally, transmucosal implant components need to promote soft tissue sealing but must not enhance osteoconductive properties. In this context, the unfilled PMMA resulted in the better induction of osteogenetic genes than that of its composites, supporting the potential use of the latter for the intended application.

## 4. Conclusions

In this work, three different PMMA/Al_2_O_3_ composites, as well as an unfilled PMMA control, were prepared by adding to the polymer matrix 30, 40, and 50 wt.% of Al_2_O_3_, respectively, with the aim of producing good candidates suitable for implant-supported prostheses. The unfilled PMMA showed a fairly smooth and compact surface, while the alumina particles in the composites appeared to be not well embedded in the matrix and caused the formation of holes. As these effects were proportionally more evident when increasing the Al_2_O_3_ content, PMMA/Al_2_O_3_ 50/50 showed an extremely uneven surface with large and numerous particles agglomerates and holes as well as particles trapped within the holes. The Al_2_O_3_ particles accounted for an increase in the flexural modulus of PMMA. Specifically, the flexural modulus of the composite containing 40 wt.% Al_2_O_3_ was about 18% higher than that of the unfilled polymer. In the PMMA/Al_2_O_3_ 50/50 composites, the flexural modulus remained almost constant. In contrast, with the addition of alumina, filler aggregates and internal defects (i.e., voids) caused a decrease in the PMMA flexural strength. The surface roughness (R_a_) and water contact angle had the same trend, varying from 1.94 µm and 77.2° for the unfilled PMMA to 2.45 µm and 105.8° for PMMA/Al_2_O_3_ 50/50. The serum protein adsorption was evaluated, obtaining a statistically significant improvement in the amount of protein (almost doubled) adsorbed on the composites incorporating the highest percentage of alumina (i.e., 40 and 50 wt.%), compared with the unfilled PMMA and PMMA/Al_2_O_3_ 70/30. The adhesion pattern of the HaCaT cells reproduced that of serum protein adsorption, as PMMA/Al_2_O_3_ 70/30 did not outperform the unfilled polymer matrix; conversely, the presence of alumina significantly increased the number of adherent mesenchymal stem cells and fibroblasts compared with the unfilled PMMA. No cytotoxic effects were detected, hence confirming that all the specimens are biocompatible and capable of sustaining cell growth and proliferation, without remarkable differences at 24 and 48 h. In addition, the presence of alumina in PMMA caused strong cell responses, thus driving tissue formation in contact with the composite itself, without improving its osteoconductive properties.

Overall, these results allow us to identify the PMMA/Al_2_O_3_ 60/40 composite as the most promising solution for rapid cell adhesion when soft tissue sealing needs to be improved. The incorporation of alumina into PMMA may thus be useful not only in improving the mechanical behavior of the material but also the biological response, as this filler ameliorates the interaction of biomolecules with the substrate at least in a short investigation period. Despite the positive results of the investigated PMMA composites, there is still much work to be done before these materials may be used in dental clinics for actual applications. In fact, the dental implant market is well established, with various materials, such as titanium alloys and zirconia ceramics, having a proven track record of success. Therefore, for new materials to penetrate the market is not easy. Further research should focus on long-term stability, wear resistance, and in vivo performance in either simulated oral environments or in vivo clinical studies.

## Figures and Tables

**Figure 1 polymers-15-03186-f001:**
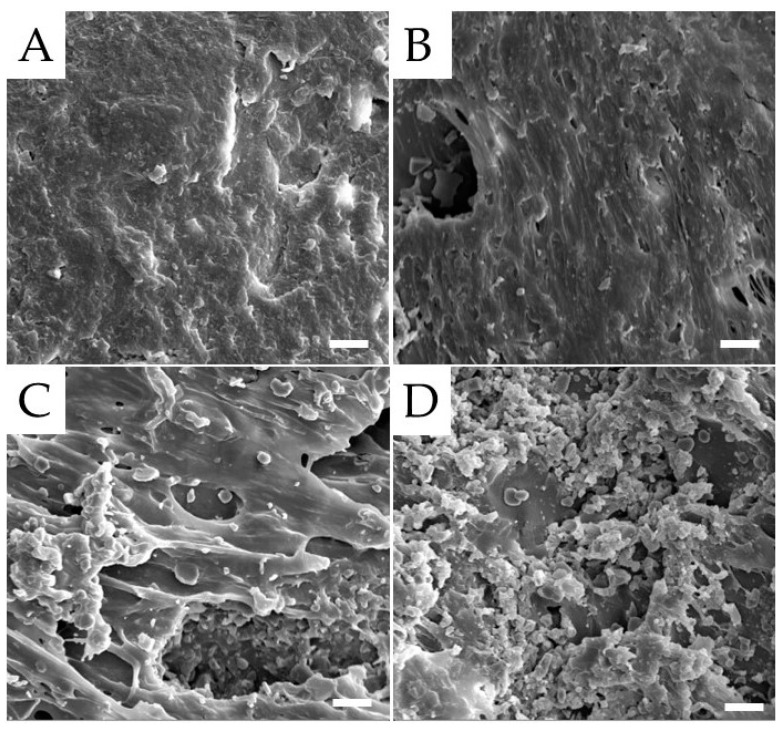
SEM micrographs of the surfaces of the samples: (**A**) unfilled PMMA; (**B**) PMMA/Al_2_O_3_ 70/30; (**C**) PMMA/Al_2_O_3_ 60/40; and (**D**) PMMA/Al_2_O_3_ 50/50. Scale bar: 10 µm.

**Figure 2 polymers-15-03186-f002:**
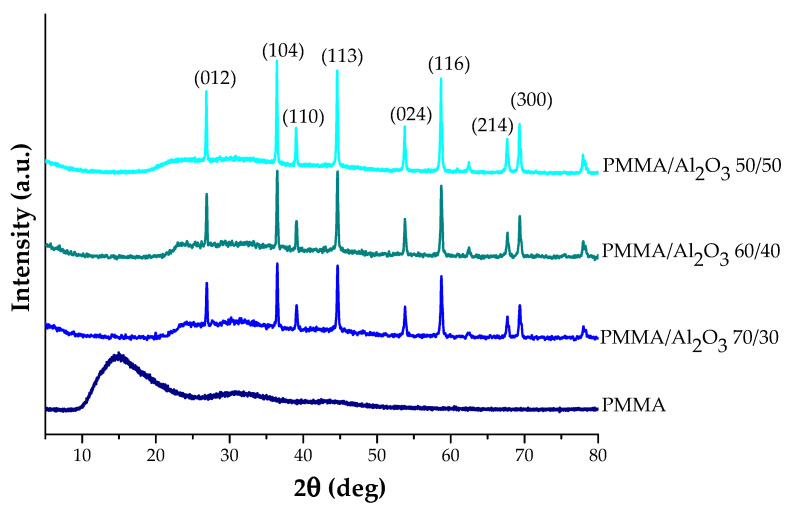
WAXD patterns of PMMA and PMMA/Al_2_O_3_ composites.

**Figure 3 polymers-15-03186-f003:**
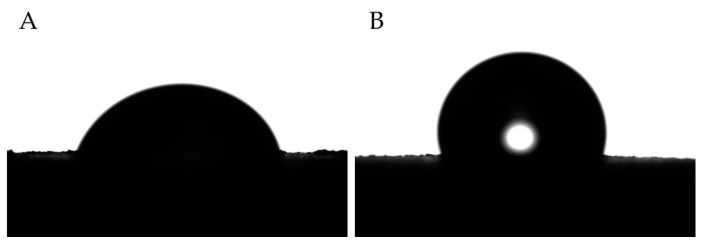
(**A**) Water droplet on PMMA with CA of 77.2 ± 3.8; (**B**) water droplet on the PMMA/Al_2_O_3_ 50/50 composites with CA of 105.8 ± 4.6.

**Figure 4 polymers-15-03186-f004:**
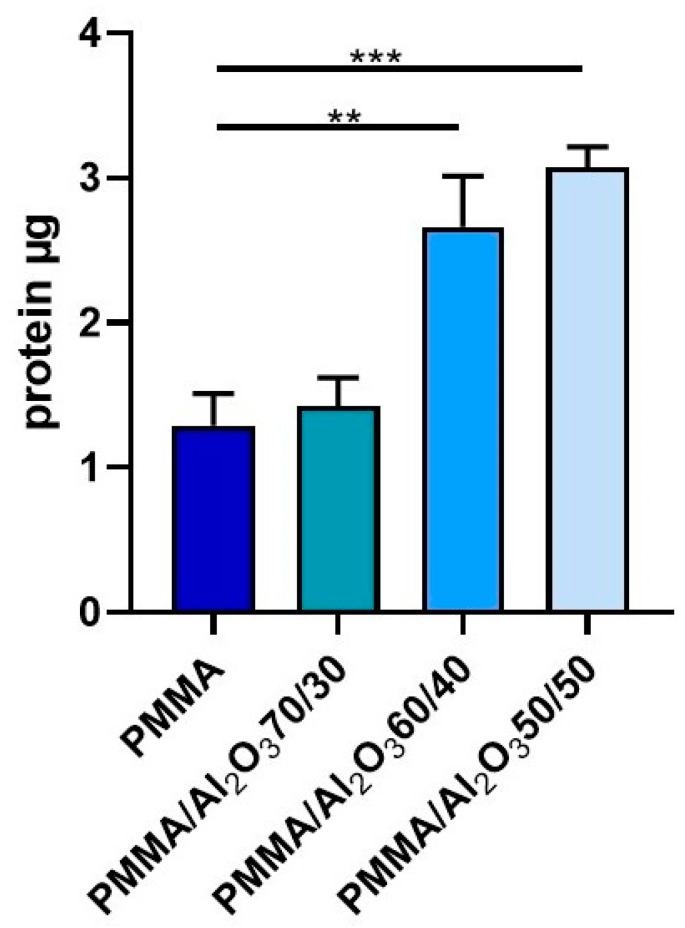
Protein adsorption with BCA-based method. Adsorbance measured at 562 nm was converted into total mg of protein adsorbed on the surface. Data represent mean ± SEM; *n* = 5. ** *p* < 0.01; *** *p* < 0.001. Statistical analysis: ordinary one-way ANOVA.

**Figure 5 polymers-15-03186-f005:**
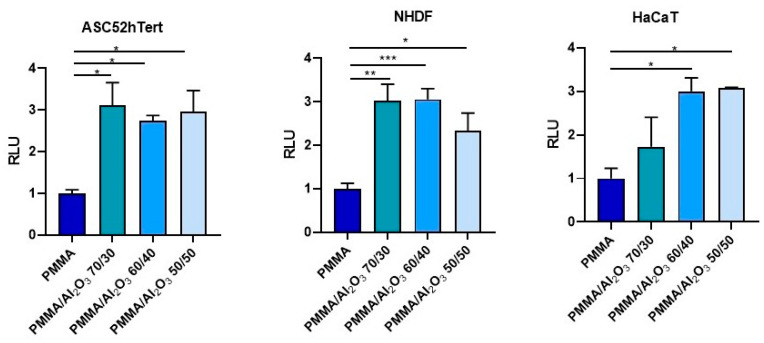
Cell adhesion assay performed in ASC (**left**), HFF (**middle**), and HaCaT (**right**) with an ATP-based method at 15 min. Data are normalized versus control (PMMA). Data represent mean ± SEM; *n* = 5; * *p* < 0.05; ** *p* < 0.01; *** *p* < 0.001. Statistical analysis: ordinary one-way ANOVA (RLU = relative light unit; ASC = adipose stem cells; HFF = human fibroblasts; HaCaT = keratinocytes.

**Figure 6 polymers-15-03186-f006:**
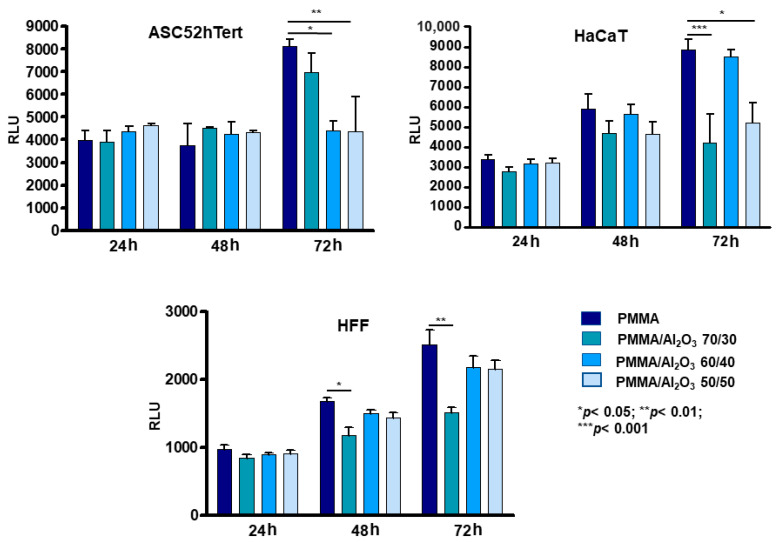
Cell viability in ASC (**top right**), HaCaT (**top left**) and HFF (**down**) with an ATP-based method after 24, 48, and 72 h of culture. Data represent mean ± SEM; *n* = 5; * *p* < 0.05; ** *p* < 0.01; *** *p* < 0.001. Statistical analysis: ordinary one-way ANOVA (RLU = relative light unit; ASC = adipose stem cells; HFF = human fibroblasts; HaCaT = keratinocytes.

**Figure 7 polymers-15-03186-f007:**
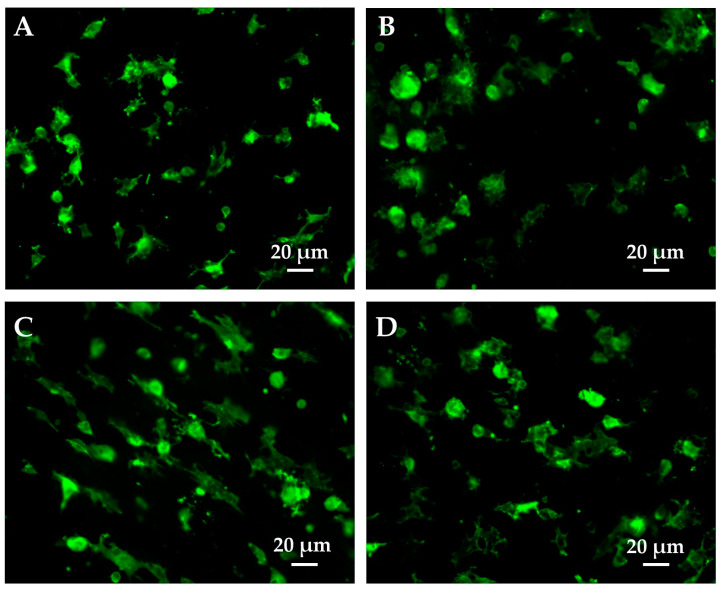
Representative images of cell morphology at 1 h after seeding in PMMA (**A**), PMMA/Al_2_O_3_ 70/30 (**B**), PMMA/Al_2_O_3_ 60/40 (**C**), and PMMA/Al_2_O_3_ 50/50 (**D**). Cell cytoskeleton was stained with phalloidin (green).

**Figure 8 polymers-15-03186-f008:**
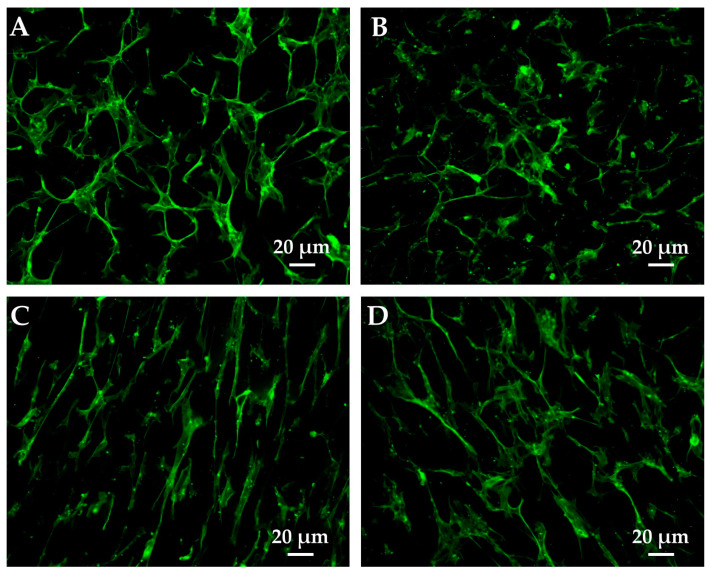
Representative images of cell morphology at 24 h after seeding in PMMA (**A**), PMMA/Al_2_O_3_ 70/30 (**B**), PMMA/Al_2_O_3_ 60/40 (**C**), and PMMA/Al2O3 50/50 (**D**). Cell cytoskeleton was stained with phalloidin (green).

**Figure 9 polymers-15-03186-f009:**
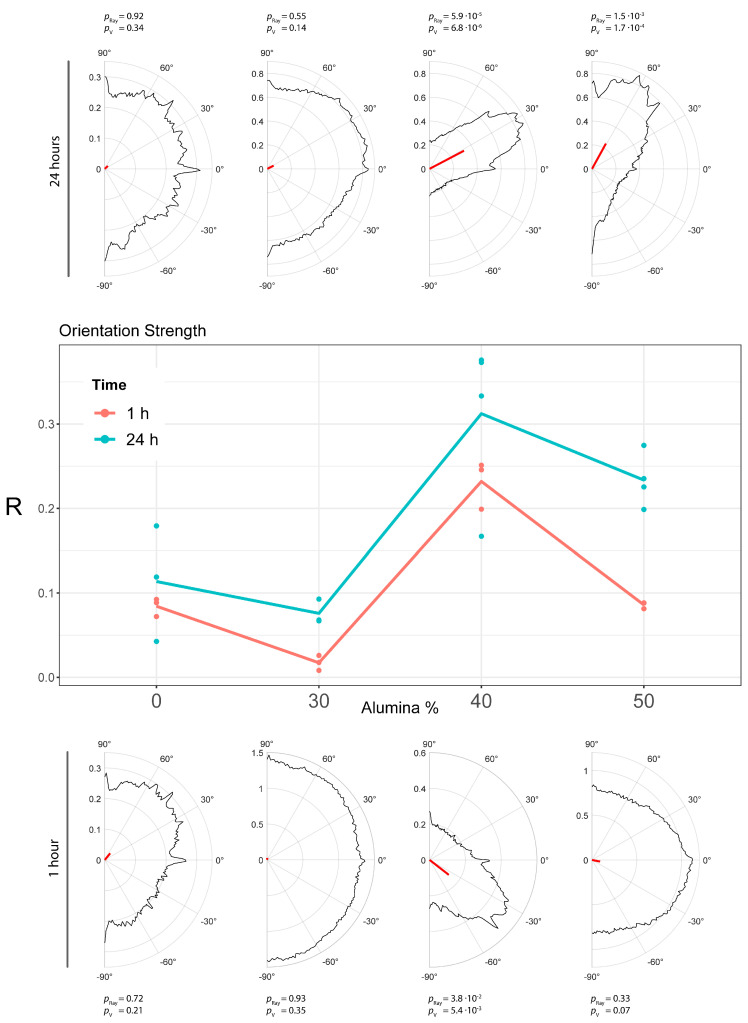
Orientation strength analysis at 1 and 24 h. (**Top**): half-circle distributions with R in red for PMMA and PMMA composites at 24 h. (**Middle**): Line graph, correlation between R and wt.% of alumina. (**Bottom**): half-circle distributions with R in red for PMMA and PMMA composites at 1 h.

**Figure 10 polymers-15-03186-f010:**
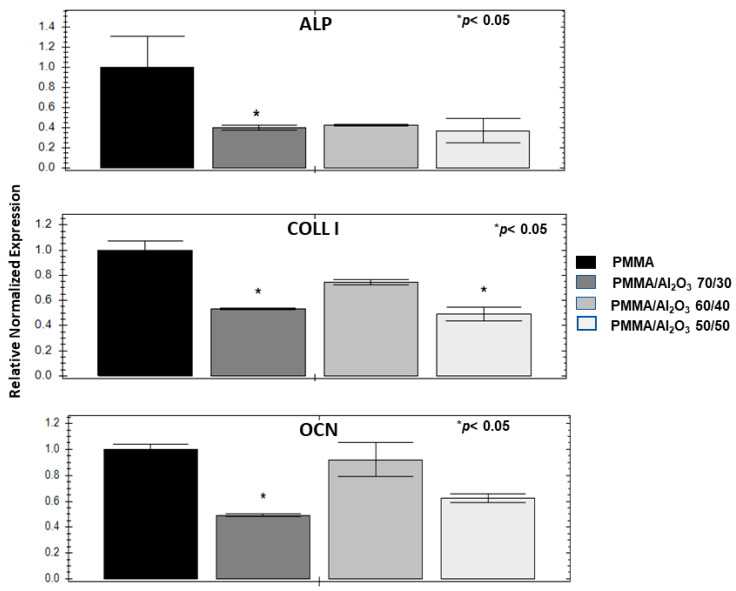
Osteoconductive assessment on ASCs: charts of the gene expression of alkaline phoshatase (ALP), collagen type I (COLL I), and osteocalcin (OCN) expressed by ASCs cultured in osteogenic medium for 30 days; SEM images of ASCs maintained at the same condition.

**Figure 11 polymers-15-03186-f011:**
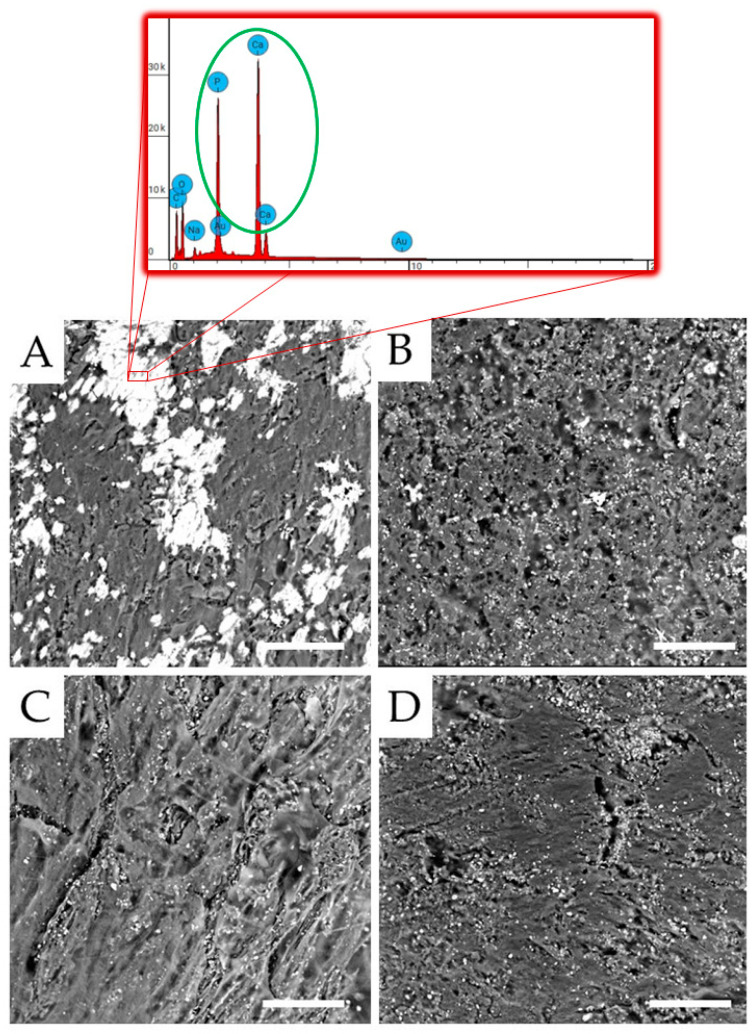
SEM images of ASCs cultured in osteogenic medium for 30 days for PMMA (**A**), PMMA/Al_2_O_3_ 70/30 (**B**), PMMA/Al_2_O_3_ 60/40 (**C**), and PMMA/Al_2_O_3_ 50/50 (**D**). Scale bar: 100 µm. Inset: calcium content by EDX.

**Table 1 polymers-15-03186-t001:** Hardness, flexural parameter, water contact angle, and microroughness of PMMA/Al_2_O_3_ composites.

	Hardness (Shore D)	Flexural Modulus (GPa)	Flexural Strength (MPa)	Elongation at Break (%)	Water Contact Angle (°)	Roughness (µm)
PMMA	91.8 ± 1.5	3.86 ± 0.80	104.2 ± 15.5	3.21 ± 0.63	77.2 ± 3.8	1.94 ± 0.33
PMMA/Al_2_O_3_ 70/30	96.8 ± 1.2	4.39 ± 0.26	80.5 ± 4.6	2.04 ± 0.12	99.7 ± 5.3	2.28 ± 0.35
PMMA/Al_2_O_3_ 60/40	96.8 ± 1.6	4.50 ± 0.13	72.1 ± 3.7	1.78 ± 0.06	103.6 ± 10.0	2.32 ± 0.39
PMMA/Al_2_O_3_ 50/50	97.0 ± 1.2	4.44 ± 0.13	65.4 ± 3.2	1.64 ± 0.08	105.8 ± 4.6	2.45 ± 0.44

**Table 2 polymers-15-03186-t002:** Orientation analysis raw data from MORPHEUS software. Mean = mean degree of orientation, R = mean resultant vector of the distribution, S = standard deviation (1 − R), *p*-values of Rayleigh test and V-test. In each SEM picture, there were 50–200 cells.

Row	Mean	R	S	Rayleigh_Pval	Vtest_Pval
PMMA_0A_1_1	35.67701600	0.072035935	0.927964065	0.474441214	0.110761126
PMMA_0A_1_2	50.63379808	0.088506355	0.911493645	0.721981796	0.208632835
PMMA_0A_1_3	52.86819961	0.092353823	0.907646177	0.701346336	0.198654567
PMMA_0A_24_1	82.74359410	0.118870528	0.881129472	0.352655849	0.074071439
PMMA_0A_24_2	42.39168138	0.042559193	0.957440807	0.921683113	0.342375407
PMMA_0A_24_3	73.89992077	0.179232651	0.820767349	0.136471136	0.022975108
PMMA_30A_1_1	−61.48795269	0.025929487	0.974070513	0.965958569	0.395724331
PMMA_30A_1_2	−68.56930945	0.008236939	0.991763061	0.996173616	0.464959623
PMMA_30A_1_3	86.88319288	0.017617649	0.982382351	0.927777572	0.349160372
PMMA_30A_24_1	28.58235944	0.066870283	0.933129717	0.527206483	0.128637695
PMMA_30A_24_2	26.84561799	0.067942743	0.932057257	0.548369580	0.136178979
PMMA_30A_24_3	43.90423995	0.092727566	0.907272434	0.625224085	0.165392251
PMMA_40A_1_1	−1214876817	0.199050932	0.800949068	0.002521487	0.000282729
PMMA_40A_1_2	−26.82576711	0.245658642	0.754341358	0.021715219	0.002912159
PMMA_40A_1_3	−37.88083450	0.251121755	0.748878245	0.038089695	0.005407698
PMMA_40A_24_1	72.53467140	0.375807448	0.624192552	7.21462 × 10^−7^	8.27644 × 10^−8^
PMMA_40A_24_2	79.77644706	0.333230014	0.666769986	0.000127143	1.40318 × 10^−5^
PMMA_40A_24_3	60.54074263	0.166938425	0.833061575	0.097362791	0.015490374
PMMA_40A_24_4	27.69148361	0.372972822	0.627027178	5.90978 × 10^−5^	6.82014 × 10^−6^
PMMA_50A_1_1	−4.806702539	0.088296361	0.911703639	0.663567024	0.181657465
PMMA_50A_1_2	−17.28843119	0.087947123	0.912052877	0.647910275	0.174922305
PMMA_50A_1_3	−11.75231262	0.081283560	0.918716440	0.337785021	0.070193658
PMMA_50A_24_1	−61.55226548	0.235267368	0.764732632	0.005880955	0.000708117
PMMA_50A_24_2	73.61116062	0.225430377	0.774569623	0.006611798	0.000799675
PMMA_50A_24_3	73.74353041	0.198685652	0.801314348	0.008193347	0.000998012
PMMA_50A_24_4	61.23645716	0.274637545	0.725362455	0.001507763	0.000171249

## Data Availability

Not applicable.

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
