# Peer review of "Mechanical and Biological Characterization of PMMA/Al2O3 Composites for Dental Implant Abutments"

_polymers, 2023, doi:10.3390/polym15153186_

Round 1
Reviewer 1 Report
1、The abstract and introduction sections need improvement, especially the logic of language.
2、In the Introduction, it is mentioned that PMMA/Al2O3 composites were successfully prepared 20 years ago. Has it been applied in the field of dental implants? What are the major problems that hindered the clinical application of PMMA/Al2O3 composites?
3、What are the advantages and differences of PMMA/Al2O3 composites compared to ATZ?
4、The Introduction mentioned that the compatibility between Al2O3 and the polymer matrix is poor, and they tend to aggregate. However, in this study, a relatively high content (30%-50%) of Al2O3 was used, which resulted in aggregation. How was the Al2O3 content used in this study selected and determined? Would a lower content yield better results?
5、What is the reason for the increase in viscosity caused by alumina in Section 3.1? Has it been experimentally validated?
6、In Section 3.2, what are the required mechanical properties for dental implants? Can the PMMA/Al2O3 composites prepared in this study meet those requirements?
7、Improve the style of Table 2 to match Table 1.
8、The styles of the scale bar should be consistent for all figures and the scale should be appropriate.
9、The English language of the manuscript needs to be polished.
10、The image resolution of the figures should be improved.
The English language of the manuscript needs to be polished.
Reviewer 2 Report
In this work, the authors reported the “Mechanical and biological characterization of PMMA/Al2O3 composites for dental implant abutments”. However, this manuscript must improve in some ways before being accepted in Polymers.
1. The abstract should be written more precisely without including unnecessary information. Try to highlight the novelty of the research and its contribution accurately with numerical results.
2. the chemical structure of PMMA/Al2O3 composites should be confirmed.
3. how about wide-angle XRD patterns for the samples?
4. One critical aspect to consider is the stability of the materials, particularly for practical applications.
5. The scale bar is not visible in the SEM images.
6. The WAC images of the samples should be provided.
7. The reviewer suggests that the authors include their perspective on the future of this research in the conclusion section.
Moderate editing of English language required
Round 2
Reviewer 2 Report
The manuscript is okay now for publication.